# Zero-Shot Image Classification Method Based on Attention Mechanism and Semantic Information Fusion

**DOI:** 10.3390/s23042311

**Published:** 2023-02-19

**Authors:** Yaru Wang, Lilong Feng, Xiaoke Song, Dawei Xu, Yongjie Zhai

**Affiliations:** 1Department of Automation, North China Electric Power University, Baoding 071003, China; 2State Key Laboratory of Management and Control for Complex Systems, Institute of Automation, Chinese Academy of Sciences, Beijing 100190, China

**Keywords:** image classification, attention mechanism, matrix decomposition, attributes, word vectors

## Abstract

The zero-shot image classification (ZSIC) is designed to solve the classification problem when the sample is very small, or the category is missing. A common method is to use attribute or word vectors as a priori category features (auxiliary information) and complete the domain transfer from training of seen classes to recognition of unseen classes by building a mapping between image features and a priori category features. However, feature extraction of the whole image lacks discrimination, and the amount of information of single attribute features or word vector features of categories is insufficient, which makes the matching degree between image features and prior class features not high and affects the accuracy of the ZSIC model. To this end, a spatial attention mechanism is designed, and an image feature extraction module based on this attention mechanism is constructed to screen critical features with discrimination. A semantic information fusion method based on matrix decomposition is proposed, which first decomposes the attribute features and then fuses them with the extracted word vector features of a dataset to achieve information expansion. Through the above two improvement measures, the classification accuracy of the ZSIC model for unseen images is improved. The experimental results on public datasets verify the effect and superiority of the proposed methods.

## 1. Introduction

In recent years, deep learning algorithms have made rapid progress in the image recognition field, but they require significant human and material resources to obtain a sufficient quantity of manually annotated data [1]. In many practical applications, a large quantity of labeled data is difficult to obtain, and the variety of objects is increasing, which requires the computer training process to constantly add new samples and new object types [2,3]. The problem of how to use computers and existing knowledge to classify and identify samples with insufficient or even completely missing label data has become a pressing problem. For this reason, ZSIC [4] was created. It is a technique that trains a learning model to predict and recognize data without class labels (unseen classes) based on some sample data with class labels (seen classes), supplemented by relevant common-sense information or a priori knowledge (auxiliary information) [5,6].

To achieve ZSIC, a popular strategy is to learn the mapping or embedding between the semantic space of classes and the visual space of images based on seen classes and the semantic description of each category. Semantic descriptions of categories usually include attributes [7], word vectors [8], gaze [9], and sentences [10]. At present, the embedded-based methods [11,12,13,14,15] are used to learn visual-to-semantic, semantic-to-visual, or latent intermedium space, so that visual and semantic embedding can be compared in shared space. Then, the unseen classes are classified by nearest neighbor search.

Most of the existing embedding methods, either based on end-to-end convolution neural networks or deep features, emphasize learning the embedding between global visual features and semantic vectors, which leads to two problems [16]. First, there are only slight differences between some features of seen and unseen classes. For some datasets, the inter-class difference is even smaller than the intra-class. Therefore, global image features cannot effectively represent fine-grained information, which is difficult to distinguish in semantic space. Second, compared to visual information, semantic information is not rich enough. The attribute features of categories are usually based on manual annotation, rely on professional knowledge, and are limited by the dimension of visual cognition. The dimension of attribute features is usually not high, and as intermediate auxiliary information, the amount of information is insufficient [17]. The word vectors are mostly obtained through models such as word2vec [18], GloVe [19], or fastText [20]. Relatively speaking, the word vectors may contain more noise and are difficult to combine with human prior knowledge; thus, their interpretability and discriminability are poor. Therefore, the imbalanced supervision from the semantic and visual space can make the learned mapping easily overfitting to seen classes. Inspired by the attention mechanism in the field of natural language processing, a few methods [16,21,22,23] introduce attention thinking into ZSIC. These methods learn regional embedding of different attributes or similarity measures based on attribute prototypes and learn to distinguish partial features, but they ignore the global features and the information imbalance of semantic and visual space.

Based on the above observation, this paper proposes an improved ZSIC model. The main contributions are as follows:(1)A feature attention mechanism is designed, and an image feature extraction module based on the attention mechanism is built. The features in different regions of the image are assigned attention weights to distinguish the key and non-key local features, and then the local features are fused with the global features.(2)A semantic information fusion module based on matrix decomposition is built. The matrix decomposition method is used to transform the binary features of attributes into continuous features and transform their dimensions to be the same as word vectors. In addition, attribute features are fused with word vector features to obtain more accurate and richer fused semantic features as a priori category features.(3)The improved ZSIC model promotes the alignment of semantic information and visual features. Experiments on the public dataset show that the improved ZSIC model improves image classification accuracy.

## 2. Related Work

### 2.1. ZSIC Methods

Recent ZSIC methods focus on learning better visual–semantic embeddings. The core idea is to learn a mapping between the visual and attribute/semantic domains and transfer semantic knowledge from seen to unseen classes according to the similarity measure. Some methods [11,12,24,25] follow the visual-to-semantic mapping direction and align visual features and semantic information in semantic space. However, when high-dimensional visual features are mapped to a low-dimensional semantic space, the shrink of feature space would aggravate the hubness problem [26,27] that in some instances in the high-dimensional space becomes the nearest neighbors of a large number of instances. To tackle these problems, some methods [13,14,28,29,30] map semantic embedding to visual space and treat the projected results as class prototypes. Shigeto et al. [31] experimentally proved that the semantic-to-visual embedding is able to generate more compact and separative visual feature distribution with the one-to-many correspondence manner, thereby mitigating the hubness issue. Ji et al. [32] also follow the inverse mapping direction from semantic space to visual space and proposed a semantic-guided class imbalance learning model which alleviates the class-imbalance issue in ZSIC. In addition, for the class-imbalance issue, the generative models have been introduced to learn semantic-to-visual mapping to generate visual features of unseen classes [33,34,35,36,37] for data augmentation. Currently, the generative ZSIC is usually based on variational autoencoders (VAEs) [37], generative adversarial nets (GANs) [33], and generative flows [34]. However, the performance of this type of method greatly depends on the quality of generated visual features or images, which is difficult to guarantee, and the mode is prone to mode collapse. Furthermore, to alleviate the hubness issue, common space learning is also employed to learn a common representation space for interaction between visual and semantic domains [15,38,39]. However, these embedded-based models only use the global feature representation, ignoring the fine-grained details in the image, and the training results are not satisfied for the poorly identified features.

### 2.2. Attention Mechanism

The concept of attention was first introduced into natural language processing tasks. In particular, because soft attention is differentiable and can learn parameters by backpropagation of the model, it has been widely used and developed in computer vision tasks. Zhu et al. [40] applied an attention mechanism in the facial expression recognition task and proposed a cascade attention-based recognition network by a hybrid of the spatial attention mechanism and pyramid feature to improve the accuracy of facial expression recognition under uneven illumination or partial occlusion. Sun et al. and Liu et al. applied an attention mechanism in the semantic segmentation task of remote sensing images. They proposed a multi-attention-based UNet [41] and an attention-based residual encoder [42], respectively. Through channel attention and spatial attention, the capability of fine-grained features was improved. The above attention mechanism includes (i) feature aggregation and (ii) a combination of channel attention (global attention) and spatial attention (local attention), which are common branches of the attention mechanism. In addition, Obeso et al. [43] proved that the global and local attention mechanism in deep neural networks works well with the human visual attention mechanism. Inspired by the above works, several researchers incorporated an attention mechanism into models for ZSIC. For example, Yang et al. [16] proposed a semantic-aligned reinforced attention model to discover invariable features related to class-level semantic attributes from variable intra-class vision information, and thereby to avoid misalignment between visual information and semantic representations. Xu et al. [21] jointly learned discriminative global and local features using only class-level attributes to improve the attribute localization ability of image representation. Chen et al. [22] proposed an attribute-guided transformer network to enhance discriminative attribute localization by reducing the relative geometry relationships among the grid features. Yang et al. [23] proposed to learn prototypes via placeholders and proposed semantic-oriented fine-tuning for preliminary visual–semantic alignment. These methods locate salient regions according to semantic attributes and ignore meaningless information to promote the alignment between a visual space and a semantic space. Compared with these methods, we also consider the combination of local features and global features, as well as the imbalance of information in semantic and visual space.

## 3. Materials and Methods

The basic embedding-based ZSIC model framework is shown in Figure 1.

The image feature extraction layer uses a deep CNN to extract image features and input them to a middle embedding layer. A priori class information (auxiliary information) is usually attribute features or word vector features. In the middle embedding layer, the correlation between image features and a priori class information is calculated. Let the total number of seen classes be *n* and a priori class feature vector of the *i*-th seen class be $\beta_{i}$, whose dimension is *m*. In the training stage of the model, the images $x_{i}$ belonging to the *i*-th seen class are input into the image feature extraction layer to extract *m*-dimensional image feature vectors $\;\alpha_{x_{i}}$; $\;\alpha_{x_{i}}$ and $\beta_{i}$ are input into the middle embedding layer, and a relationship similarity $\left(\alpha_{x_{i}},\beta_{i}\right)$ between $\;\alpha_{x_{i}}$ and $\beta_{i}$ is established to obtain the matching score. Cosine distance is used to calculate the matching score. Compared with the European distance, cosine distance is more consistent with the distance calculation form of the high-dimensional vector, and its formula is
(1)$$\mathrm{score}=\mathrm{similarity}\left(\alpha_{x_{i}},\beta_{i}\right)=\frac{\sum_{k=1}^{m}a_{k}b_{k}}{\sqrt{\sum_{k=1}^{m}a_{k}{}^{2}}\sqrt{\sum_{k=1}^{m}b_{k}{}^{2}}}$$
where $\;\alpha_{x_{i}}=\left[a_{1},a_{2},\ldots ,a_{m}\right]$ and $\beta_{i}=\left[b_{1},b_{2},\ldots ,b_{m}\right]$.

In order to match the image feature vectors and the prior class feature vectors belonging to the same class as closely as possible, that is, to maximize the matching score, the loss function is used as follows:(2)$$\mathrm{loss}=-\frac{1}{n}\sum_{i=1}^{n}\frac{\;\alpha_{x_{i}}\cdot \beta_{i}}{\left\|\;\alpha_{x_{i}}\|\cdot \|\beta_{i}\right\|}$$

In the testing stage of the model, the image feature vectors of unseen classes are extracted through the feature extraction layer and then matched with the prior class feature vectors corresponding to each class in the middle embedding layer. When the matching score is the highest, the corresponding class is the prediction class of the input image.

Using the above model framework, the improved embedding-based ZSIC model is shown in Figure 2. Details are as follows.

### 3.1. IFE-AM Module

In ZSIC tasks, image features need to be matched with a priori class features, while image features extracted by CNN correspond to a whole image, so they lack discrimination. Therefore, an image feature extraction module based on an attention mechanism (IFE-AM) is constructed (as shown in Figure 2) to focus high-level image features on the key regions of the input image, in order to reduce the deviation from the priori class features and improve the degree of matching. The typical convolutional neural networks VGG-19 and ResNet-34 are taken as examples to illustrate the attention mechanism designed in this paper.

The flowchart of the spatial attention mechanism that weights the feature vector of each position is shown in Figure 3.

Let the output features of the last layer of the CNN be ***F***, with dimension [*x*, *y*, *p*], which contains *p* channels. For ***F***, set window [*x*, *y*], and use max pooling and average pooling to obtain two *p*-dimensional feature vectors $F_{\max}$ and $F_{\mathrm{mean}}$, respectively, and then concatenate them to obtain $\left[\;F_{\max},\;F_{\mathrm{mean}}\right]$. Then, $\left[\;F_{\max},\;F_{\mathrm{mean}}\right]\;$ is connected to the fully connected (FC) layer, the hidden layer unit is set as *p*, and a *p*-dimensional query vector ***Q*** is output for feature selection of the attention mechanism. The feature map of the *i*-th channel in ***F*** is recorded as ***f****_i_*, *i* = 1, 2, …, *p*, and its size is *x* × *y*; the feature vector of the *j*-th position in ***F*** is recorded as ***l****_j_*, *j* = 1, 2, …, *x* × *y*, and its size is *p* × 1. Calculate the dot product of ***Q*** and ***l****_j_* to obtain the feature weight *w_j_* of the *j*-th position, and then use the softmax function for normalization to obtain the feature weight matrix ***W***. The formula is as follows:(3)$$W=\mathrm{softmax}\left(w_{j})=\mathrm{softmax}(\mathrm{dot}(Q^{T},l_{j}\right))$$

The feature values at different positions in ***f****_i_* are weighted and summed according to the weight matrix ***W***, and $F_{\mathrm{attention}}$ is output.

Finally, based on the idea of residual connection, the feature vectors $F_{\max}$, $F_{\mathrm{mean}}$**,** and $F_{\mathrm{attention}}$ are summed to obtain the final output eigenvector $F_{\mathrm{output}}$.

### 3.2. SIF-MD Module

ZSIC methods rely on prior class information to complete the transfer from seen classes to unseen classes, so accurate and informative class description information is the key. Currently, the commonly used a priori class description information includes attribute features and word vector features. In order to make the two types of a priori class description information complementary and improve the amount of information, a semantic information fusion module based on matrix decomposition (SIF-MD) is constructed, as shown in Figure 2.

Usually, the dimensions of manually set attribute information is small, and the attribute features are all binary features of 0 or 1, which are relatively sparse and independent; the dimensions of word vectors are relatively large, which are characterized by continuity between [–1, 1]. To carry out information fusion, the matrix decomposition method is used to transform the binary features of attributes into continuous features and transform their dimensions to be the same as word vectors. The architecture diagram of the matrix decomposition of attributes is shown in Figure 4.

First, use attribute matrix ***D*** (*M* × *N*) to represent *n*-dimensional attribute vectors of m classes, which is decomposed into U (*M* × *K*) and ***V*** (*N* × *K*) with the equation
(4)$$D=UV^{T}$$
where k  is the dimension of the matrix decomposition. Make $UV^{T}$ as close as possible to ***D***, that is, fitting attribute feature ***D*** through matrix U  and matrix V. The loss function is the mean squared error MSE (mean squared error) method:(5)$$\mathrm{loss}=\overset{M}{\underset{i=1}{\sum }}\overset{N}{\underset{j=1}{\sum }}{\left(D_{i,j}-{\hat{D}}_{i,j}\right)}^{2}$$
(6)$${\hat{D}}_{i,j}=U_{i}V_{j}^{T}$$
where $U_{i}$ denotes the vector in the *i*-th row of matrix U, *i* = 1, 2, …, *M*, and $V_{j}$ denotes the vector in the *j*-th row of matrix V, *j* = 1, 2, …, *N*.

To prevent overfitting, the L2 canonical term is added to Formula (5):(7)$$\mathrm{loss}=\overset{m}{\underset{i=1}{\sum }}\overset{n}{\underset{j=1}{\sum }}{\left(D_{i,j}-{\hat{D}}_{i,j}\right)}^{2}+\lambda \left(\|U_{i}{\|}_{1}+\|V_{j}{\|}_{1}\right)$$

Each row in U is a *k*-dimension vector, which matches the dimension of the word vector of the corresponding class. The matrix U and the word vector matrix W(*m* × *k*) are summed in certain weight proportions as fused semantic features $W_{\mathrm{add}}$, which are given by
(8)$$W_{\mathrm{add}}=\alpha W+\left(1-\alpha \right)U$$
where α is a parameter with a range of [0, 1]; $W_{\mathrm{add}}$ is a fused semantic feature, retaining the content of attribute features and word vector features.

## 4. Experiment Results

The experiment is based on the 4× 1080Ti GPU server of Ubuntu16.04, the Python 3.6 virtual environment is built through Anaconda, and deep learning frameworks of TensorFlow1.2.0 and Keras2.0.6 are installed.

The top-1 accuracy and top-3 accuracy were used to evaluate the classification results of the zero-shot classification model on the test set. The training set and test set were randomly selected four times to obtain four groups of experimental results, and the average classification accuracy was recorded.

### 4.1. Dataset

The experiment was conducted based on the Animals with Attributes 2 (AwA2) [27] dataset. AwA2 is a public dataset for attribute-based classification and zero-shot learning, and it is publicly available at http://cvml.ist.ac.at/AwA2, accessed on 9 June 2017. The dataset contains 37,322 images and 50 animal classes, and each class has an 85-dimensional attribute vector. It is a coarse-grained dataset that is medium-scale in terms of the number of images and small-scale in terms of the number of classes. In experiments, we followed the standard zero-shot split proposed in reference [9], that is, 40 classes for training and 10 classes for testing. The training set and test set do not intersect. Among the training set, 13 classes were randomly selected for validation to perform a hyperparameter search.

### 4.2. Ablation Experiment of IFE-AM Model

According to the model structure shown in Figure 2, the experiments were conducted with the representative VGG-19 and ResNet-34 as the backbone networks, which are called VGG-A and ResNet-A, respectively. The image features were extracted by the pre-improved and improved networks, and the attribute features of the dataset were used to conduct experiments.

#### 4.2.1. Training Loss and Classification Accuracy

When the model is trained, the training loss is calculated according to Formula (2). Figure 5 shows the change curves of the training loss (train_loss) corresponding to different feature extraction networks.

Table 1 shows the epochs required for training and train_loss values corresponding to different feature extraction networks, as well as the classification accuracy (top-1 and top-3) of the test set.

Figure 5 and Table 1 show that the train_loss of the ResNet-34 model decreases faster than the VGG-19 model. The final train_loss of the VGG-19 and ResNet-34 models tends to be stable, but the train_loss of the ResNet-34 model is lower. From the decreasing trend in train_loss, the train_loss of the VGG-19 model fluctuates greatly, and the decreasing process of train_loss of the ResNet-34 model is more stable. The ResNet-A model is also superior to the VGG-A model in decreasing speed and the stability of train_loss. This shows that the ResNet-34 model with residual connections can realize matching between image features and prior class features faster, better, and more stably. In addition, for both the VGG-A model and ResNet-A model, although their train_loss overall declines slightly slower, their required training epoch and loss value after stabilization are significantly lower than those of the original VGG-19 and ResNet-34 networks. This shows that the IFE-AM module proposed in this paper, as a feature-weighted focusing strategy, improves the model’s ability to capture image features in space, thus realizing further fitting of deep features; additionally, the attention mechanism is based on the method of weighted information fusion, which makes the acquisition and update of information more stable, thus achieving a faster and more stable fitting effect.

For the image classification results of the test set, the top-1 and top-3 of the ResNet-34 model are all larger than those of the VGG-19 model, which shows that its residual structure has a good effect on the fitting of deep image features. The top-1 and top-3 of the ResNet-A model are higher than those of the VGG-19 and ResNet-34 models without the attention mechanism, which shows that the attention mechanism can focus the features of spatial attention and effectively improve the generation of image features and the matching effect with prior class features. The accuracies of VGG-A and ResNet-A are similar, but the top-3 of ResNet-A is significantly improved, which shows that the ResNet-A model can obtain more accurate image features in high-dimensional space, making the distance between classes farther, the distance within classes closer, and the matching effect with semantic features better.

#### 4.2.2. Feature Segmentation

According to the model shown in Figure 4, for VGG-A and ResNet-A, the image feature $F_{\mathrm{output}}=F_{\max}+F_{\mathrm{mean}}+F_{\mathrm{attention}}$ is split, and $F_{\max}$, $F_{\mathrm{mean}}$ and $F_{\mathrm{attention}}$ are, respectively, output to the next layer for comparison with $F_{\mathrm{output}}$. The accuracy of the final image classification is shown in Table 2 and Table 3.

As shown in Table 2 and Table 3, the image classification results of the improved ResNet-A model based on the attention mechanism are better than those of the VGG-A model. Whether it is the VGG-A or ResNet-A model, the image classification accuracy corresponding to different image features satisfies $F_{\mathrm{output}}>F_{\mathrm{attention}}>F_{\mathrm{mean}}>F_{\max}$, which verifies the effect of image feature extraction based on the spatial attention mechanism. Inspired by the idea of residual connection, the three features are superposed to obtain $F_{\mathrm{output}}$, which fuses the information of different features and finally obtains the optimal image classification result.

### 4.3. Ablation Experiment of SIF-MD Module

Since the above experiments verified that ResNet-A and $F_{\mathrm{output}}$ are better, the following further experiments are conducted on these bases. Three models of word2vec, GloVe, and fastText were used to extract the word vector features of each class in the dataset, with a dimension of 256. The attribute features of the dataset were decomposed according to Formulas (4)–(7), and the loss threshold value was set as 0.1. Then, the decomposed attributes were weighted and fused with word vector features extracted by word2vec, GloVe, and fastText, respectively, according to Formula (8). The fusion parameter α was set as [0, 1] and the step size as 0.1.

The image classification experiment of the test set was repeated five times, and the average value of the top-1 was taken. The experimental results corresponding to different word vectors and different fusion parameters α are shown in Table 4. Figure 6 more intuitively shows the changing trend of top-1 accuracy with α when different word vectors are used as auxiliary information.

As shown in Figure 6, the top-1 accuracy of the word vector extracted by GloVe as prior class features is significantly higher than that extracted by word2vec or fastText. As shown in Table 4, when α = 0, that is, only the attribute features are used as the prior class feature, the top-1 accuracy of image classification is 43.1%. When α = 1, that is, only word vectors are used as prior class features, the top-1 accuracies corresponding to word2vec and GloVe are 44.2% and 44.7%, respectively, which are better than the results when only attribute features are used, while the top-1 accuracy corresponding to fastText is lower than the results when only attribute features are used. For the word vectors extracted by word2vec, GloVe, and fastText, the fusions with attribute feature all have positive effects. For the word2vec word vector, when the fusion weight α = 0.8 and 0.9, the top-1 accuracy is 1.2% and 1.4% higher than that of the attribute vector only and 0.1% and 0.3% higher than that of the word vector only, respectively. For the fastText word vector, when the fusion weight α = 0.2, 0.3, and 0.4, the top-1 accuracy is 0.2%, 0.5%, and 0.1% higher than that of the attribute vector only and 1.2%, 1.5%, and 1.1% higher than that of the word vector only, respectively. For the GloVe word vector, when the fusion weight α = 0.6, 0.7, 0.8, and 0.9, the top-1 accuracy is 1.9%, 2.0%, 2.7%, and 2.2% higher than that of the attribute vector only and 0.3%, 0.4%, 1.1%, and 0.6% higher than that of the word vector only, respectively. The results show that it is meaningful to fuse attribute features and the word vector features.

## 5. Discussions

To verify the effectiveness of the method proposed, the method is compared with the baseline model and existing classical models. The baseline model only uses the deep learning network ResNet-34 or VGG-19 to extract image features and uses attributes or word vectors as auxiliary information. The results of the comparative experiment are shown in Table 5 and Figure 7. In the table, “ResNet-34 + attribute” refers to the model that uses ResNet-34 to extract image features and uses attributes as auxiliary information. The image classification results were evaluated with top-1 accuracy. The experimental results of IAP, CONSE, and CMT adopt the results given in references [27,31]. The dataset and the splits of the training set and test set in the experiments of all methods are the same as that of our method, and no methods were pre-trained by large datasets (such as ImageNet).

As shown in Table 5 and Figure 7, for the baseline model, the top-1 accuracy of the model using ResNet-34 to extract image features is higher than that of the model using the VGG-19 network; the top-1 accuracy of the model using word vectors extracted by word2vec or GloVe as auxiliary information is higher than that of the model using attributes; and the top-1 accuracy of the “ResNet-34 + GloVe” method is the highest, with a value of 42.7%. The top-1 accuracy of our method is 3.1% higher than that of the “ResNet-34 + GloVe” method. For existing classical methods, IAP detects unseen classes based on attribute transfer between classes, the attribute features are limited by the dimension of visual cognition, and the amount of information is insufficient. CONSE uses CNN to extract image features without distinguishing the importance of different regional features, and only uses word vectors extracted by word2vec as auxiliary information. CMT uses Sparse Coding to extract image features and uses a neural network architecture to learn the word vectors of categories. Although more semantic word representations are learned by using local and global contexts, the discrimination of word vectors is poor, and the imbalanced supervision between semantic features and visual features is still large. Our method assigns attention weights to different regions of the image through the SIF-MD module and strengthens the key features highly related to semantic information. In addition, it alleviates the imbalanced supervision issue between semantic features and visual features through IFE-AM module. These improvements promote the alignment of visual features and semantic information and make the matching degree of the two higher, which is very important for ZSIC. Thus, the top-1 accuracy of our method is 9.9% higher than IAP, 1.3% higher than CONSE, and 7.9% higher than CMT. The above experimental results prove the effectiveness of our method.

## 6. Conclusions

To improve the accuracy of the ZSIC model based on embedded space, the IFE-AM model and SIF-MD module are constructed in this paper. After the existing CNN is used to extract the image feature map, the max pooling, average pooling, and spatial attention methods are used to obtain three feature vectors, and then they are fused as the final image features. The attribute matrix of the dataset is decomposed to match its dimensions with the extracted word vector, and then the attribute and word vector are weighted and fused as auxiliary information of the improved ZSIC model.

Experiments were conducted on a public dataset. First, the ablation experiment of the IFE-AM model was carried out. The experimental results show that the top-1 and top-3 accuracies corresponding to ResNet-A are 1.6% and 7.8% higher than those of ResNet-34, respectively; the top-1 and top-3 accuracies corresponding to VGG-A are 3.1% and 7.8% higher than those of VGG-19, respectively. Then, the ablation experiment of the SIF-MD module was carried out. The experimental results show that the top-1 accuracies of using fused semantic information as auxiliary information are significantly higher than that of using attribute or word vector alone. Third, comparative experiments were carried out, and the results show that the accuracy of the proposed method is significantly higher than the baseline method and several existing classical methods.

For different types of semantic information, the fusion parameter is not fixed and needs to be determined by experiments. How to derive the value of the fusion parameter in theory is our future work. A small- to medium-sized dataset is considered in our work, and larger data scenarios will be explored in the future.

## Figures and Tables

**Figure 1 sensors-23-02311-f001:**
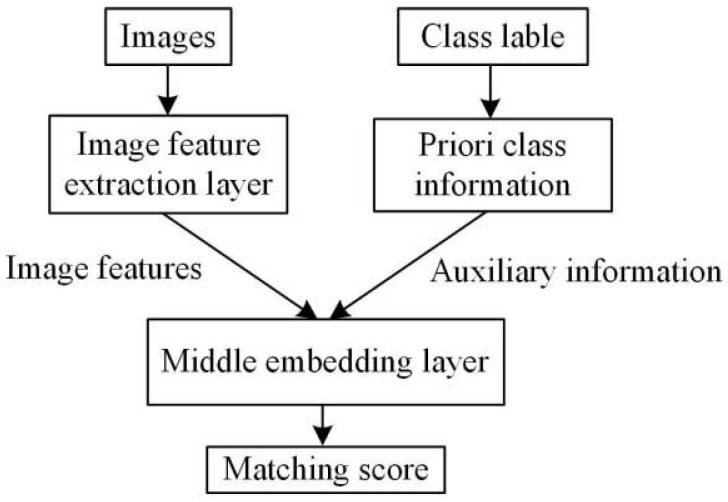
Basic embedding-based ZSIC model framework.

**Figure 2 sensors-23-02311-f002:**
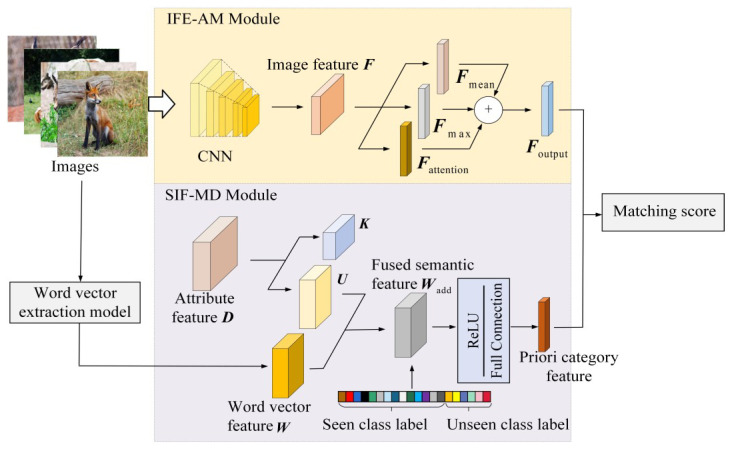
Improved ZSIC model.

**Figure 3 sensors-23-02311-f003:**
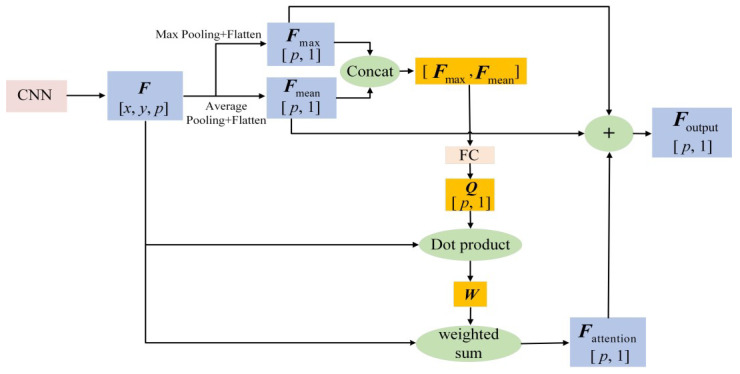
Flowchart of the attention mechanism.

**Figure 4 sensors-23-02311-f004:**
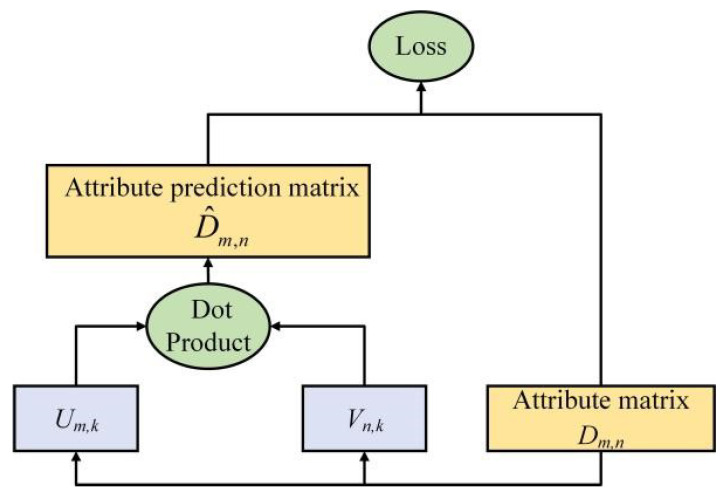
Architecture diagram of the matrix decomposition of attributes.

**Figure 5 sensors-23-02311-f005:**
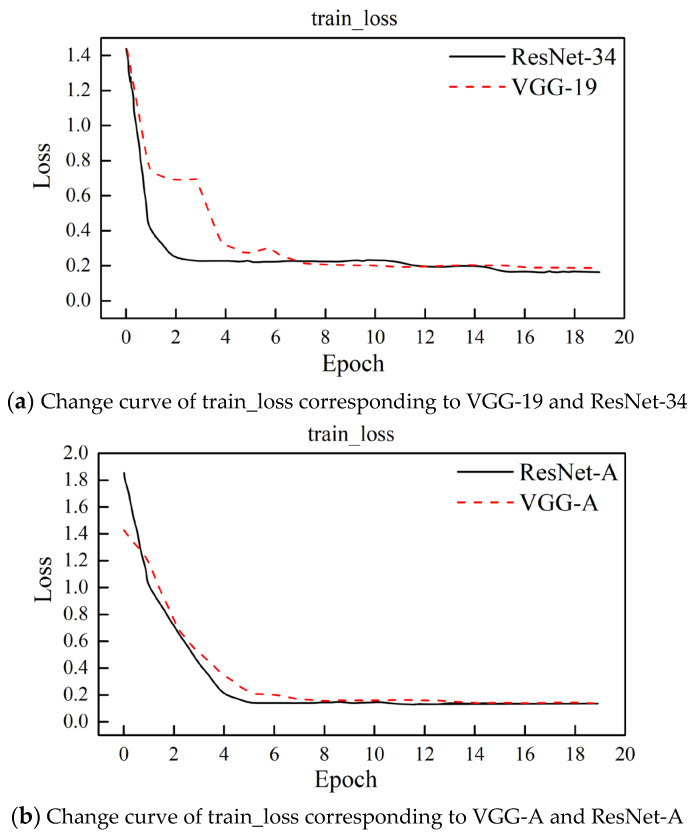
Change curves of train_loss.

**Figure 6 sensors-23-02311-f006:**
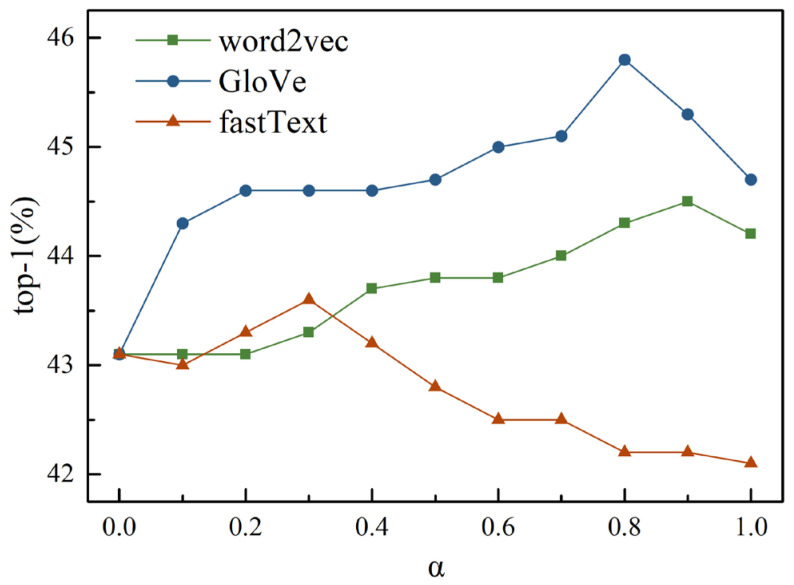
Changing trend of top-1 accuracy of image classification.

**Figure 7 sensors-23-02311-f007:**
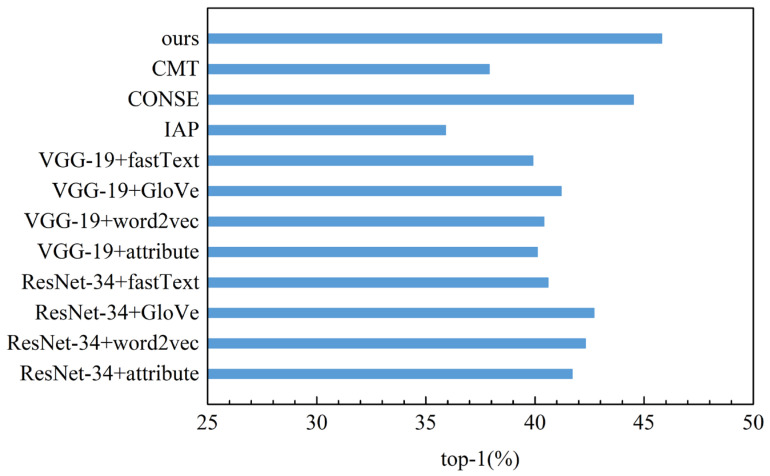
Top-1 accuracy comparison of different methods.

**Table 1 sensors-23-02311-t001:** Test results.

Feature Extraction Network	IFE-AM	Epochs	Train_Loss	Top-1 (%)	Top-3 (%)
VGG-19		17	0.174	40.1	53.1
ResNet-34		16	0.155	41.7	56.1
VGG-A	√	13	0.147	43.2	60.9
ResNet-A	√	5	0.139	43.3	63.9

**Table 2 sensors-23-02311-t002:** Comparison of different image features in the VGG-A model.

Image Features	Attention	Feature Fusion	Top-1 (%)	Top-3 (%)
$F_{\max}$			39.9	45.0
$F_{\mathrm{mean}}$			40.3	51.1
$F_{\mathrm{attention}}$	√		40.9	51.9
$F_{\mathrm{output}}$	√	√	42.3	60.9

**Table 3 sensors-23-02311-t003:** Comparison of different image features in the ResNet-A model.

Image Features	Attention	Feature Fusion	Top-1 (%)	Top-3 (%)
$F_{\max}$			39.1	41.1
$F_{\mathrm{mean}}$			41.7	56.1
$F_{\mathrm{attention}}$	√		42.9	61.1
$F_{\mathrm{output}}$	√	√	43.3	63.9

**Table 4 sensors-23-02311-t004:** Image classification top-1 accuracy of the test set.

Word Vector	α										
	0	0.1	0.2	0.3	0.4	0.5	0.6	0.7	0.8	0.9	1
word2vec	43.1	43.1	43.1	43.3	43.7	43.8	43.8	44.0	44.3	44.5	44.2
GloVe	43.1	44.3	44.6	44.6	44.6	44.7	45.0	45.1	45.8	45.3	44.7
fastText	43.1	43.0	43.3	43.6	43.2	42.8	42.5	42.5	42.2	42.2	42.1

**Table 5 sensors-23-02311-t005:** Image classification results of different methods.

	Method	Top-1 (%)
1	ResNet-34 + attribute	41.7
2	ResNet-34 + word2vec	42.3
3	ResNet-34 + GloVe	42.7
4	ResNet-34 + fastText	40.6
5	VGG-19 + attribute	40.1
6	VGG-19 + word2vec	40.4
7	VGG-19 + GloVe	41.2
8	VGG-19 + fastText	39.9
9	IAP	35.9
10	CONSE	44.5
11	CMT	37.9
12	ours	45.8

## Data Availability

Not applicable.

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
