# Peer review of "Zero-Shot Image Classification Method Based on Attention Mechanism and Semantic Information Fusion"

_sensors, 2023, doi:10.3390/s23042311_

Round 1
Reviewer 1 Report
This paper proposes an improved zero-shot image classification model, including IFE-AM Module and SIF-MD Module, which improves the accuracy of image classification. It is a meaningful work, but there are the following questions:
1. Attributes and word vectors belong to different types of semantic information. Whether to consider the dimension mismatch when merging attributes and word vectors? How is it solved?
2. In the model shown in Figure 2, how to calculate the matching score between image features and semantic features?
3. On Line 151 of the text, it is written as “Calculate the dot product of query vector Q and the p-dimensional feature vector of each position in the feature map F to obtain the feature weights of different positions.” However, Figure 3 shows the dot product operation between CNN and Q matrix. How to understand this calculation?
4. In Line 89 of the text, the innovation and main work introduction of this article are relatively simple, and it is recommended to expand this part appropriately.
5. Some highly related works are suggested for discuss, such as: Semantic-Guided Class Imbalance Learning Model for Zero-Shot.
6. What needs to be further improved in the future work?
Author Response
This paper proposes an improved zero-shot image classification model, including IFE-AM Module and SIF-MD Module, which improves the accuracy of image classification. It is a meaningful work, but there are the following questions:
1、 Attributes and word vectors belong to different types of semantic information. Whether to consider the dimension mismatch when merging attributes and word vectors? How is it solved?
Response: Thank you for your comments. Attributes and word vectors really belong to different types of semantic information. And we considered the possible dimension mismatch issue. The matrix decomposition method is used to transform the dimensions of attributes to be the same as word vectors. Relevant contents are in Line223-235 of the manuscript.
2、In the model shown in Figure 2, how to calculate the matching score between image features and semantic features?
Response: Thank you for your comments. Cosine distance is used to calculate the matching score. Calculation method are in Line158-163 of the manuscript.
3、 On Line 151 of the text, it is written as “Calculate the dot product of query vector Q and the p-dimensional feature vector of each position in the feature map F to obtain the feature weights of different positions.” However, Figure 3 shows the dot product operation between CNN and Q matrix. How to understand this calculation?
Response: We are sorry. Figure 3 is wrong. This line between “CNN” and "Dot product" should go from "F [x, y, p]". This figure has been modified. Relevant explanations are in Line188-205 of the manuscript.
4、In Line 89 of the text, the innovation and main introduction of this article are relatively simple, and it is recommended to expand this part appropriately.
Response: Thank you for your comments. This part has been appropriately expanded in Line70-83 of the manuscript.
5、Some highly related works are suggested for discuss, such as: Semantic-Guided Class Imbalance Learning Model for Zero-Shot.
Response: Thank you for your comments. The work of this reference is excellent, and we have discussed this reference in Line 98-101 of the manuscript. In addition, we have also discussed many other relevant documents, in Line85-143 of the manuscript.
6、What needs to be further improved in the future work?
Response: Thank you for your comments. The work needs to be improved in the future is written in the last paragraph of the article. For different types of semantic information, the fusion parameter are not fixed and need to be determined by experiments. How to derive the value of the fusion parameter in theory is our future work. Small to medium-sized dataset is considered in our work, larger data scenarios will be explored in the future.
Reviewer 2 Report
The paper applies Zero-shot learning and attention mechanism to the research of image classification, which has certain practical value in the field of Cv. However, the paper should also address the following issues before publication,
1) There are two main problems in the introduction section of the paper. First of all, the review of the literature is too superficial. Taking Line47-77 as an example, although the author listed several related studies, he did not systematically sort out the connections and progressive relationships between these works and only used the listing method to conduct a literature review. It will make the reader feel that the logic is not strong. Secondly, as an important part of the manuscript, the review of the attention mechanism is not enough. At the same time, the references in this paper are a bit old. It is recommended that the author supplement some latest literature, such as the following, but not limited to this .
https://doi.org/10.3390/s22041350
https://doi.org/10.3390/sym14050906
https://doi.org/10.3390/rs14133109
https://doi.org/10.1016/j.patcog.2021.108411
2) This paper was not well organized. The Relate Work in the paper is more like Methodology. It is recommended that the author merge this part t into the next chapter.
3) In addition, the introduction of the data set in this article is not comprehensive. Line200-209 seems to be an independent research data chapter. The author may consider separating the content of this subsection.
4) Discussion is recommended to be an independent chapter. The current paper lacks real Discussion in essence. The author should introduce more about the applicability and limitations of the model, or compare it with similar works. At present, there are few relevant introductions.
5) This is a paper on image classification, but there are no related figures/images in the full text, which will also make readers feel unreasonable.
Author Response
The paper applies Zero-shot learning and attention mechanism to the research of image classification, which has certain practical value in the field of Cv. However, the paper should also address the following issues before publication,
1、There are two main problems in the introduction section of the paper. First of all, the review of the literature is too superficial. Taking Line47-77 as an example, although the author listed several related studies, he did not systematically sort out the connections and progressive relationships between these works and only used the listing method to conduct a literature review. It will make the reader feel that the logic is not strong. Secondly, as an important part of the manuscript, the review of the attention mechanism is not enough. At the same time, the references in this paper are a bit old. It is recommended that the author supplement some latest literature, such as the following, but not limited to this .
https://doi.org/10.3390/s22041350
https://doi.org/10.3390/sym14050906
https://doi.org/10.3390/rs14133109
https://doi.org/10.1016/j.patcog.2021.108411
Response: Thank you for your valuable comments. The sections of “Introduction” and “Related Work” have all been rewritten. Discussions on the related literature are wroten in the section of “Related Work”. A large number of recent literature have been supplemented and some old ones have been replaced. We discussed the motivation of different methods proposed, the problems that can be solved, and the problems that still exist. The logical relationship between the literature is enhanced. At the same time, in the “Related Work” section, a review of some literature related to attention mechanism has been added, including not only the literature you put forward but also other relevant literature. Moreover, it points out the novelty of the proposed method in this paper compared with the existing methods.
2、This paper was not well organized. The related work in the paper is more like Methodology. It is recommended that the author merge this part into the next chapter.
Response: Thank you for this constructive suggestion. The content of the original “Related Work” section has been moved to the next chapter and this section has been rewritten.
3、In addition, the introduction of the data set in this article is not comprehensive. Line200-209 seems to be an independent research data chapter. The author may consider separating the content of this subsection.
Response: Thank you for this suggestion. We have added an introduction to the dataset as an independent section in “4.1 Dataset”. The contents are as follows:
The experiment is conducted based on the Animals with Attributes 2 (AwA2) [27] dataset. AwA2 is a public dataset for attribute-based classification and zero-shot learning, and it is publicly available at http://cvml.ist.ac.at/AwA2. The dataset contains 37,322 images and 50 animal classes, and each class has an 85-dimensional attribute vector. It is a coarse-grained dataset that is medium-scale in terms of the number of images and small-scale in terms of the number of classes. In experiments, we followed the standard zero-shot split proposed in reference[9], that is, 40 classes for training and 10 classes for testing. The training set and test set do not intersect. Among the training set, 13 classes were randomly selected for validation to perform a hyperparameter search.
It should be noted that this is a commonly used public dataset in the field of zero-shot image classification. There is only the above content on its official website (http://cvml.ist.ac.at/AwA2). Users can freely download the images in the dataset. In fact, the introduction of this dataset in other literature on zero-shot image classification is similar. Generally, the introduction is usually considered sufficient because this article does not analyze and study the data of this dataset, but only explains the dataset used in this experiment and how the training, validation, and test sets are divided. The following sections of this paper will use the proposed method to classify the images of this dataset, and the accuracy of image classification is the focus of this article。
4、Discussion is recommended to be an independent chapter. The current paper lacks real Discussion in essence. The author should introduce more about the applicability and limitations of the model, or compare it with similar works. At present, there are few relevant introductions.
Response: Thanks for your valuable comment. We have added a “Discussions” section to compare the method proposed in this paper with the baseline model and other classical methods of the same kind. The experimental results have been detailedly explained. Please see Discussions section.
5、This is a paper on image classification, but there are no related figures/images in the full text, which will also make readers feel unreasonable.
Response: Thank you for this valuable comment. In section 4.3 of the revised manuscript, we have added Figure 6 which shows the changing trend of top-1 accuracy with α when different word vectors are used as auxiliary information. Among them, word vectors are extracted from word2vec, GloVe, and fastText models, respectively. Through this figure, we can see the contrast more intuitively. In the Disccusion section, we added Figure 7 to intuitively show the accuracy of image classification by different methods, which is convenient for readers to compare.
Reviewer 3 Report
In the section “Results and discussion”, the authors mostly describe the obtained results while the discussion of the results is limited. For examples, a comparison of the results of this study with other similar studies should be made and an explanation of the differences in the statistical significant.
More details about training, testing and verification data should be explained
Author Response
1、In the section “Results and discussion”, the authors mostly describe the obtained results while the discussion of the results is limited. For examples, a comparison of the results of this study with other similar studies should be made and an explanation of the differences in the statistical significant.
Response: Thanks for your valuable comment. We have added a “Discussions” section to compare the method proposed in this paper with the baseline model and other classical methods of the same kind. The experimental results have been detailedly explained. Please see “Discussions” section.
2、More details about training, testing and verification data should be explained.
Response: Thank you for this constructive comment. In Line256-260, we have added an explanation of the division of the training, testing, and verification data. In experiments, we followed the standard zero-shot split proposed in reference[9], that is, 40 classes for training and 10 classes for testing. The training set and test set do not intersect. Among the training set, 13 classes were randomly selected as verification data to perform a hyperparameter search. This is the standard division commonly used in the field of zero-sample image classification, which is convenient for comparison with other algorithms.
Reviewer 4 Report
1. Please also cite previous studies based on idea related to the feature attention mechanism and mention the novelty and superiority of the proposed method compared to those. For example, the following paper may be relevant.
Zhu, Yizhe, et al. "Semantic-guided multi-attention localization for zero-shot learning." Advances in Neural Information Processing Systems 32 (2019).
2. "Dot product" in Figure 3 is not a simple dot product (in Equation (3)), so another explanation would be better. Also, the line from CNN to "Dot product" would be more understandable if it were changed to a line from "F[x,y,p]".
3. What formula is used to calculate train_loss in Figure 5? It seems to be different from the loss in Figure 4, but it is confusing.
Please check and correct the following incorrect expressions and variable notations.
4. In line 147, there are two F's that are not in bold italic.
5. For Equations (4) through (8), the subscripts representing the matrix size (number of rows and columns) and the subscripts representing the location of the matrix components (row and column) are mixed and not easy to understand. It is not common practice to use subscripts to denote the matrix size. In addition, some scalar quantities, such as Di,j and α, are incorrectly expressed in bold.
6. For equation (6), the middle equality is wrong and unnecessary. k is already defined as the size of the matrix.
7. For the right hand side of equation (7), the first term is a scalar, but the second term is a vector quantity, which seems to be wrong.
Author Response
1、Please also cite previous studies based on idea related to the feature attention mechanism and mention the novelty of the proposed method compared to those. For example, the following paper may be relevant.
Zhu, Yizhe, et al. "Semantic-guided multi-attention localization for zero-shot learning." Advances in Neural Information Processing Systems 32 (2019).
Response: Thank you for your valuable comments. The sections of “Introduction” and “Related Work” have all been rewritten. Discussions on the related literature are wroten in the section of “Related Work”. A large number of recent literature have been supplemented and some old ones have been replaced. The literature you put forward is excellent, and we have cited this literature in Line 130-133. In addition, some other relevant literature related to attention mechanism has been cited. And it points out the novelty of the proposed method in this paper compared with the existing methods, in Line 139-143.
2、"Dot product" in Figure 3 is not a simple dot product (in Equation (3)), so another explanation would be better. Also, the line from CNN to "Dot product" would be more understandable if it were changed to a line from "F[x,y,p]".
Response: Thank you for the mistakes you pointed out. this picture is wrong. This line should really go from "F [x, y, p]" to the dot product. Figure 3 has been modified and improved, and a more detailed explanation has been added, in Line 188-205.
3、What formula is used to calculate train_loss in Figure 5? It seems to be different from the loss in Figure 4, but it is confusing.
Response: Thank you for your comments. Sorry, we didn't write it clearly. A description was added in Line 268 of the manuscript. The train_loss in Figure 5 is calculated by formula (2), unlike the loss in Figure 4, and the loss in Figure 4 is only used for SIF-MD Module.
Please check and correct the following incorrect expressions and variable notations.
4、In line 147, there are two F's that are not in bold italic.
Response: Thank you for the mistakes you pointed out. All the above problems have been corrected, on Line 193.
5、For Equations (4) through (8), the subscripts representing the matrix size (number of rows and columns) and the subscripts representing the location of the matrix components (row and column) are mixed and not easy to understand. It is not common practice to use subscripts to denote the matrix size. In addition, some scalar quantities, such as Di,j and α, are incorrectly expressed in bold.
Response: Thank you for the mistakes you pointed out. All the above problems have been corrected, on Line 223-241.
6、For equation (6), the middle equality is wrong and unnecessary. k is already defined as the size of the matrix.
Response: Thank you for pointing out the error. In the original manuscript, the symbol indicating the size of the matrix and the subscript are confused. On Line 234 of the revised manuscript, the middle quality has been deleted.
7、For equation (7), the first term is a scalar, but the second term is a vector quantity, which seems to be wrong.
Response: Thank you for pointing out the error. In the original manuscript, the symbol representing the matrix row and the symbol representing the matrix column are confused. The error has been corrected in the revised manuscript, On Line 234. For the right hand side of equation (7), the first and second term are all scalar。
Round 2
Reviewer 2 Report
The authors made a detailed explanation or modification of each problem. Now the quality of the paper has been significantly improved. I accept the paper published in the Sensors journal.
Reviewer 3 Report
I think the paper can be accepted in the current form